# Effectiveness and limitations of parameter tuning in reducing biases of top-of-atmosphere radiation and clouds in MIROC version 5

**Tomoo Ogura[1], Hideo Shiogama[1], Masahiro Watanabe[2], Masakazu Yoshimori[3], Tokuta Yokohata[1], James D. Annan[4], Julia C. Hargreaves[4], Naoto Ushigami[5], Kazuya Hirota[2], Yu Someya[2], Youichi Kamae[5], Hiroaki Tatebe[6], and Masahide Kimoto[2]**

[1] National Institute for Environmental Studies, Tsukuba, Ibaraki, Japan

[2] Atmosphere and Ocean Research Institute, University of Tokyo, Kashiwa, Chiba, Japan

[3] Faculty of Environmental Earth Science, Global Institution for Collaborative Research and Education, and Arctic Research Center, Hokkaido University, Sapporo, Hokkaido, Japan

[4] BlueSkiesResearch.org.uk, Settle, North Yorkshire, United Kingdom

[5] University of Tsukuba, Tsukuba, Ibaraki, Japan

[6] Japan Agency for Marine-Earth Science and Technology, Yokohama, Kanagawa, Japan

Correspondence to: T. Ogura (ogura@nies.go.jp)

**Abstract**

This study discusses how much of the biases in top-of-atmosphere (TOA) radiation and clouds can be removed by parameter tuning in the present-day simulation of a climate model in the Coupled Model Inter-comparison Project phase 5 (CMIP5) generation. We used output of a perturbed parameter ensemble (PPE) experiment conducted with an Atmosphere-Ocean General Circulation Model (AOGCM) without flux adjustment. The Model for Interdisciplinary Research on Climate version 5 (MIROC5) was used for the PPE experiment. Output of the PPE was compared with satellite observation data to evaluate the model biases and the parametric uncertainty of the biases with respect to TOA radiation and clouds. The results indicate that removing or changing the sign of the biases by parameter tuning alone is difficult. Especially, the cooling bias of the shortwave cloud radiative effect in low latitudes

could not be removed, neither in the zonal mean nor at each latitude–longitude grid point. The bias was related to the overestimation of both cloud amount and cloud optical thickness, which could not be removed by the parameter tuning either. However, they could be alleviated by tuning parameters such as the maximum cumulus updraft velocity at the cloud base. On the other hand, the bias of the shortwave cloud radiative effect in the Arctic was sensitive to parameter tuning. It could be removed by tuning such parameters as albedo of ice and snow both in the zonal mean and at each grid point. The obtained results illustrate the benefit of PPE experiments which provide useful information regarding effectiveness and limitations of parameter tuning. Implementing a shallow convection parameterization is suggested as a potential measure to alleviate the biases in radiation and clouds.

## 1   Introduction

The climate models used in Coupled Model Inter-comparison Project phase 5 (CMIP5) still exhibit significant biases in simulating present-day top-of-atmosphere (TOA) radiation, as in CMIP3 (Flato et al. 2013). The biases are especially large in the component of the shortwave cloud radiative effect (SCRE), namely the difference in shortwave radiation between all-sky and clear-sky values. The SCRE represents the radiative effect of clouds, which cool the climate system by reflecting shortwave radiation. Compared with satellite observations, however, the cooling effect of the SCRE tends to be overestimated over low-latitude oceans and underestimated over the Southern Ocean, suggesting that the models still have difficulties in simulating clouds in these regions (Nam et al. 2012, Bodas-Salcedo et al. 2014). Previous studies suggest that such biases in radiation and clouds might affect the simulated climate in remote regions or distort the cloud feedback in future projections (Trenberth and Fasullo 2010, Ceppi et al. 2012). Therefore, alleviating the biases by developing climate models is important.

There are two factors, which might contribute to the biases in climate simulated by the models: (a) inappropriate model structures, namely, equations representing the physical processes or spatial resolution of the model; and (b) inappropriate parameter values, which are specified in the equations. We therefore attempt to alleviate the biases by modifying the factors (a) and (b) within the plausible range during the model development process.

How much of the existing biases can be explained by the second factor (b)? In other words, how much of the biases can be removed by modifying only specified parameter values

(parameter tuning)? This issue is important when discussing the model development strategy because it helps to decide which factor, (a) or (b), should be given a priority to efficiently reduce the biases. If the biases in question can be completely explained by factor (b), the priority for parameter tuning would be high. In this case, removing the biases is relatively simple because parameter tuning is generally much easier than modifying the model structures. On the contrary, if most of the biases cannot be explained by factor (b), modifying model structures should be given a high priority.

A Perturbed Parameter Ensemble (PPE) experiment with a climate model is useful when discussing the above issue. In the PPE experiment, we can create different versions of a climate model in a systematic and comprehensive way by modifying the specified parameter values in the model within a plausible range (Murphy et al. 2004). If we evaluate the biases by comparing present-day climate with observation data in each version of the PPE models, we should be able to evaluate parametric uncertainty, namely, the inter-model difference of the biases due to parameter settings. This inter-model difference would also provide a measure regarding how much of the biases can be removed by parameter tuning only.

The benefit of PPE experiments, as discussed above, has been illustrated in previous studies. For example, Zhang et al. (2012) conducted a PPE experiment with an Atmosphere General Circulation Model (AGCM) and evaluated the performance of cloud simulations compared with satellite observations over various tropical regions. The results indicate that the model performance in simulating clouds is sensitive to parameter tuning. Yokohata et al. (2012) focused on different PPE experiments conducted with an Atmosphere-Ocean GCM (AOGCM), two Atmosphere-Slab ocean GCMs (ASGCMs), and an AGCM and evaluated the model performance in simulating the cloud radiative effect at TOA compared with observations. They found that the sensitivity of the model biases to parameter tuning varies widely among different regions. In the PPEs analyzed in the study, however, the sea surface temperature (SST) bias was suppressed by applying flux adjustment at the sea surface in both AOGCM and ASGCM.

In the present study, we attempt to better understand the parametric uncertainty of TOA radiation and cloud biases by using the PPE output of an AOGCM without flux adjustment. There is an advantage in using the AOGCM without flux adjustment because climate projections in the CMIP5 Multi-Model Ensemble (MME) are conducted with AOGCMs without flux adjustment and the biases of such AOGCMs are therefore directly relevant for

future projections using CMIP5 (Flato et al. 2013). If we suppress the SST biases in the AOGCMs by applying flux adjustment, the TOA radiation and cloud biases in which we are interested might be obscured. In addition, the parametric uncertainty of the biases might be overestimated if we apply flux adjustment because it allows us to include AOGCMs with large radiative imbalance at the TOA as valid samples in the PPE, while such models are not used for future projections in the CMIP5 MME.

When evaluating biases in the simulated clouds, we use output of the Cloud Feedback Model Inter-comparison Project (CFMIP) Observation Simulator Package (COSP), which is incorporated in the AOGCM. The COSP is diagnostic software that processes the GCM outputs, such as the cloud amount, and simulates the signals that would be retrieved by satellites (Bodas-Salcedo et al. 2011). It increases the chances that the difference between the model output and observation reflects real biases in the model simulation rather than observational limitations. Therefore, COSP has been widely used in previous studies, which evaluate clouds simulated by the CMIP5 MME. The studies indicate that the optical thickness of the simulated clouds tends to be overestimated compared with the observation, as in the CMIP3 (Klein et al. 2013, Nam et al. 2012, Zhang et al. 2005). In the present study, we evaluate the parametric uncertainty of this "too thick (bright) bias" by analyzing the COSP output of the PPE experiment and discuss how much of the bias can be removed by parameter tuning only.

Section 2 describes the AOGCM, design of the PPE experiment, and observation data used for the evaluation. In Section 3, we identify the biases in the TOA radiation and discuss the parametric uncertainty of the biases. We then focus on cloud biases in Section 4 to examine if the "too thick bias" can be controlled by parameter tuning. In addition, Section 5 describes which tuning parameters are effective in controlling the TOA cloud radiative effect. In Section 6, we discuss implications and limitations of the present study, as well as the potential pathway towards model improvement. Finally, we summarize the conclusions in Section 7.

## 2   Models and Methods

### 2.1   Design of the Perturbed Parameter Ensemble

We compared the output of the PPE experiment using the AOGCM in the pre-industrial control setting with the observation to evaluate the model biases. We used the Model for

Inter-disciplinary Research on Climate version 5 (MIROC5) AOGCM. The atmospheric
component has a horizontal resolution of T42 (~2.8°) with 40 vertical levels. The ocean
component is COCO4.5 with a horizontal resolution of ~1° and 49 vertical levels in addition
to a bottom boundary layer. The model is the low-resolution version of the MIROC5
AOGCM, which is used in CMIP5 with a higher resolution of T85 (~1.4°) in the atmosphere
(Watanabe et al. 2010). We confirmed that the low-resolution version ran stably and did not
suffer from significant climate drift in the pre-industrial control experiment without flux
adjustment when the standard setting of the tuning parameters was specified. The model could
also reproduce the characteristic biases of the TOA radiation and clouds of the T85 version
used in CMIP5.
The cloud parameterization of MIROC5 employs a statistical scheme. We assume that
there is small-scale fluctuation of total water $Q_t$ within the model grid box, which is
described by a probability density function (PDF), $G(Q_t)$. We also assume that the $Q_t$
exceeding supersaturation with respect to liquid, $Q_s$, takes the form of cloud liquid. Then the
cloud cover $C$ and cloud liquid content $Q_c$ are diagnosed as the integral over the saturated
part of the grid box, as follows:
$$C = \int_{Q_s}^{\infty} G(Q_t)\,dQ_t \qquad (1), \text{ and}$$
$$\overline{Q_c} = \int_{Q_s}^{\infty} (Q_t - Q_s) \cdot G(Q_t)\,dQ_t \qquad (2).$$
Overbar denotes average over the grid box. The shape of the PDF is represented by a
triangular function. The model predicts variance and skewness of the PDF, which are affected
by cumulus convection, cloud microphysics, turbulent mixing, and advection. Details of the
cloud parameterization are described by Watanabe et al. (2009).
MIROC5 also uses a cloud microphysics parameterization following Wilson and Ballard
(1999). The parameterization predicts ice water content using physically-based tendency
terms which represents nucleation, deposition and sublimation, riming, and ice melting,
among others.
We should note that perturbing specified values of tuning parameters might increase the net
radiation imbalance at TOA when conducting PPE with an AOGCM in the pre-industrial
control setting, which leads to a gradual change in climate different from the initial state

(climate drift). Such a change would make the definition of the control climate difficult. In addition, the simulated climate might not be a valid example of pre-industrial control simulations. Applying flux adjustment at the sea surface would help to suppress the climate drift by reducing the SST biases. However, it might also cover up the biases in the TOA radiation and clouds, which are sensitive to the SST. What we need here is both stable climate and SST biases, as indicated in the CMIP5 pre-industrial control experiments. Therefore, we used the output of the PPE experiment conducted in Shiogama et al. (2012), following the Suppressed Imbalance Sampling (SIS) method, in the present study. The SIS is a method to subsample members of the PPE with a small imbalance in the TOA radiation and thus with small climate drift. This enables us to study stable climates of the PPE without applying flux adjustment. Other methods analogous to the SIS have been discussed in Jackson et al. (2012) and Yamazaki et al. (2013).

The details of the SIS method are described in Shiogama et al. (2012). For reference, we also present the summary in the following. First, we select ten tuning parameters, which are considered important to the radiative forcing of $CO_2$ doubling, climate feedback, and climate sensitivity (Table 1). The selection is based on the results of sensitivity experiments using the atmospheric component of MIROC5, which shows that perturbing the ten parameters has large impact on the radiative forcing and climate feedback compared to other tuning parameters. The selected ten parameters are related to cumulus convection, cloud, turbulence, aerosol, and land surface processes. The maximum and minimum values of the parameters are determined by expert judgement so that the parameters are within the plausible range, namely, they are consistent with the observation and current understanding of the climate system. Values of the ten parameters are then selected from the maximum to minimum ranges and randomly paired to produce 5000 samples of ten dimensional vectors, following Latin Hypercube Sampling. Each vector corresponds to a set of input values for the ten tuning parameters. We further select 56 members from the 5000 samples so that the TOA radiative imbalance of the selected members is close to that of the standard model. The selection of the 56 members is conducted with the following 3 steps: 1) we conduct a PPE experiment with MIROC5 AGCM under pre-industrial condition, in which tuning parameters are changed one at a time to the minimum and maximum values before running the AGCM for 6 years, 2) output of the PPE members are linearly interpolated to estimate the TOA radiative imbalance for the 5000 samples of the tuning parameters, and finally, 3) we select 56 members in which the TOA radiative imbalance is close to that of the standard model. The number of

subsampled members, namely 56, is determined by the computational resources available. Note that the number increased from 35 in the previous study by Shiogama et al. (2012). Finally, we create 56 members of the MIROC5 AOGCM by specifying different members of the ten dimensional vectors for the model as input values for the tuning parameters.

We ran the 56 members of the MIROC5 AOGCM for 30 years with the pre-industrial control setting and confirmed that the changes in the simulated surface air temperature from the initial state (climate drift) were small. This was expected because the TOA radiative imbalance is close to that of the standard model. Years 1–10 of the simulation were considered to be a spin-up period during which the simulated climate adjusted to the modified tuning parameters. The output from years 11–30 was averaged to make a climatology. The model biases were defined as the difference of the climatology from observation data.

The observation data used for the model evaluation originate in the period of 1983-2017 (Table 2). Therefore, the model output from the historical simulation of the same period is appropriate for comparison with the observation. However, conducting the historical simulation requires an extension for more than 150 years after the pre-industrial control simulation of 30 years. This means more than 6-fold increase in computational cost, which we are not able to cover. Therefore, we decided to use the pre-industrial control simulation as a surrogate for the historical simulation, assuming that the former reproduces the biases in the latter, regarding TOA radiation and clouds. This assumption is supported by other simulation results. For example, we compared biases in the historical simulation with those in the pre-industrial control simulation using MIROC5 with the horizontal resolution of T85 (~1.4°). We confirmed that the TOA radiation and cloud biases in the two simulations were similar to each other (not shown).

## 2.2  Observation data

Table 2 summarizes the observation data which are compared with the model output. They all are monthly mean data. We defined the model biases referring to multiple observations, namely three for TOA radiation and two for the cloud amount; therefore, the observation uncertainty can be taken into account. The biases are considered robust if they are commonly seen with respect to multiple observations. The observation data for TOA radiation are derived from CERES-EBAF (Loeb et al. 2009), ERBE-S9 (Barkstrom 1984), and ISCCP-FD (Zhang et al. 2004). The data for the cloud amount are from GCM simulator-oriented ISCCP

cloud product (Pincus et al. 2012, Rossow et al. 1996) and CALIPSO-GOCCP (Chepfer et al. 2010). The cloud amount data of the ISCCP are custom-built daytime-only monthly averages, which are available from the CFMIP-OBS website (http://climserv.ipsl.polytechnique.fr/cfmip-obs). We first referred to the observation data to calculate the monthly climatology for the period in Table 2. We then interpolated the data linearly to the horizontal resolution of T42 and used them to calculate the difference from the model output.

When evaluating biases of clouds simulated by MIROC5 AOGCM, we used the output of the satellite simulation software COSP (version 1.2.2), which was implemented in the model; COSP includes software simulating satellite observations of ISCCP (Klein and Jakob 1999; Webb et al. 2001) and CALIOP lidar (Chepfer et al. 2008). We compared the cloud amount identified by the ISCCP simulator with the GCM simulator-oriented ISCCP cloud product and the one determined with the CALIOP lidar simulator with the CALIPSO–GOCCP data. We confirmed that the ISCCP simulator was implemented properly in the MIROC5 AOGCM following Zelinka et al. (2012), which means, we calculated the total sum of the cloud amount from the ISCCP simulator for all cloud top pressure and optical thickness bins and confirmed that the sum is consistent with the "native" cloud amount identified in MIROC5 AOGCM. Note that optically thin clouds with tau < 0.3 are not included in this comparison because the available "native" cloud amount does not include such clouds.

## 3   Parametric uncertainty of the TOA radiation bias

First, we present the outline of the TOA radiation bias of the MIROC5-PPE by discussing the global annual mean values in Figure 1. The biases in the net radiation are small (Figure 1a), which means that the values of all PPE members are within the range of the three observations and near the zero net radiation with no imbalance, indicated by the dashed line. This was expected because we selected these members when designing the PPE following the SIS method. If we focus on the components of the TOA radiation, however, we notice larger biases compared with the net radiation (Figures 1b,c). The largest biases appear in the SCRE; the biases range from -11.8 W/m$^2$ to -5.8 W/m$^2$. All PPE members are more than 3.0 W/m$^2$ smaller than either one of the three observations. Therefore, parameter tuning enables us to reduce the bias from -11.8 W/m$^2$ to -5.8 W/m$^2$ by as much as 50 percent; however, we cannot totally remove it or change its sign. The shortwave clear-sky component (SWclr) also exhibits

large biases in which all PPE members are larger than either one of the three observations.
Therefore, we cannot change the sign of the bias by parameter tuning only.
We should note that the SCRE biases are negatively correlated with the LCRE biases with
the correlation coefficient of -0.82. Therefore, if we reduce the SCRE bias by making it more
positive, the LCRE bias tends to be more negative. This would reduce the LCRE bias in more
than half of the PPE members. Correlations of the SCRE biases with the biases in clear sky
components are small: -0.08 with LWclr and -0.32 with SWclr.
Next, we discuss the characteristics of the radiation bias on a smaller spatial scale, as shown
by the zonal annual mean in Figure 2. We especially focus on the cloud radiative effect,
which illustrates the biases related to clouds. The negative SCRE biases, as observed in the
global mean (Figure 1c), are mostly attributable to the biases in low latitudes (Figure 2a). In
those latitudes, all PPE members are outside the range of the three observations. Therefore,
the bias cannot be eliminated or change the sign by parameter tuning, although it can be
reduced by ~30 percent. In the Arctic, on the other hand, the inter-model difference among
the PPE members tends to be larger compared with other latitudes; hence, the observations lie
within the PPE spread. Here, the SCRE bias can be eliminated or change the sign by
parameter tuning. The biases of the Longwave Cloud Radiative Effect (LCRE) appears to be
small in most latitudes (Figure 2b). At least one of the PPE members is within the range of the
three observations in most latitudes.
The characteristics on an even smaller spatial scale are illustrated by the geographical
distribution of the annual mean cloud radiative effect biases in Figures 3a and b. We used
CERES–EBAF as the observation because it measures the radiative fluxes more directly than
the ISCCP–FD and it also has various advantages over the ERBE–S9 such as scene
identification (Wielicki et al. 1996, Loeb et al. 2009). We confirmed that similar results were
obtained when using ISCCP–FD or ERBE-S9 (not shown).
The negative SCRE bias in the low latitudes, as observed in the zonal mean plot (Figure 2a),
appears pronounced over the oceans, exceeding -40 W/m$^2$ in large areas (Figure 3a). We also
notice positive biases in middle to high latitudes over the Southern Ocean, northwestern part
of Eurasia, and northeastern part of North America. They exceed 5 W/m$^2$ in some places. On
the other hand, if we measure the parametric uncertainty of the SCRE bias using the standard
deviation among the PPE members, we notice that the uncertainty does not exceed 4 W/m$^2$ in
most areas (Figure 3c). Therefore, removing or changing the sign of the SCRE bias at each

grid point by parameter tuning only is difficult. This can be confirmed by the fractions of the PPE members, which have positive biases (Figure 3e). At each grid point, we count the number of the PPE members which have positive SCRE bias. Then we divide it by the total number of the PPE members, which is 56. The resulting fractions are plotted in the Figure 3e, so that we can see if the observation data lie within the range of the PPE spread at each grid point. In most areas of the globe, the fraction is 0 (blue) or 1 (orange), which means that observation data are outside the range of the PPE spread, or all PPE members have the same sign of the SCRE bias. In this case, parameter tuning plays only a limited role in reducing the SCRE bias; especially, the sign of the bias cannot be changed. An exception is the Arctic. Here, the SCRE bias is about 5 W/m$^2$ and the standard deviation of the bias ranges from 6–8 W/m$^2$ (Figures 3a,c). The observation data are within the range of the PPE spread. Therefore, the biases of the PPE members can be either positive or negative, which is indicated by the green and yellow colours in Figure 3e. Here, we can change the sign of the SCRE bias by parameter tuning.

The LCRE bias is smaller than the SCRE bias (Figures 3a,b). It is smaller than 20 W/m$^2$ in most areas. However, the standard deviation of the LCRE bias is even smaller (Figure 3d), less than 5 W/m$^2$, except for the limited area in the tropics. Therefore, changing the sign of the LCRE bias is difficult in most regions except for northern mid-latitudes and the Southern Pacific. This is illustrated by the fractions of the PPE members, which have positive biases (Figure 3f). They are 0 (blue) or 1 (orange) in large areas including the Arctic.

## 4    Parametric uncertainty of the cloud bias

To better understand the origin of the cloud radiative effect bias, we examine the geographical distribution of the cloud amount bias in Figure 4. In the following, we present results for the boreal summer season when the cloud amount bias is most pronounced in the Hawaiian Trade Cumulus Region, which we discuss later in this section. The cloud amount is overestimated over the Pacific and Atlantic in low latitudes (Figures 4a,b), which contributes to the negative SCRE bias, as shown in Figure 3a. The overestimation is a robust feature; it exists with respect to both ISCCP and CALIPSO observations. In addition, all members of the PPE have positive biases in those regions (Figures 4c,d). Therefore, the biases cannot be removed by parameter tuning. We should note here that the multi-model mean ISCCP cloud amount (tau > 1.3) from the CFMIP1 and CFMIP2 ensembles does not show such positive

bias in low latitudes (Klein et al. 2013). Therefore, the bias might be a problem specific to the
MIROC5 AOGCM.
The cloud amount bias can be decomposed into the contributions from different cloud top
pressure and optical thickness bins, as illustrated for the Hawaiian Trade Cumulus Regions
(15-35°N, 160°E-140°W) in Figure 5. The region of focus is indicated by the black square in
Figure 4b. The MIROC5-PPE tends to overestimate optically thick clouds (tau > 3.6) and
underestimate optically thin clouds (tau < 3.6) compared with the ISCCP observation (Figures
5a,b,c). The contribution of the former outweighs that of the latter, which leads to the
overestimation of the cloud amount. The overestimation is especially large in low-top clouds
(pc > 680). The clouds of the MIROC5-PPE are biased towards optically thick clouds
compared with the observation, which also contributes to the negative SCRE bias.
We further examined the signs of the cloud biases for each bin of the cloud top pressure and
optical thickness categories. The fraction of the positive biases within the PPE members is 0
(blue) or 1 (orange) in 36 out of 42 bins (Figure 5d); all PPE members have the same cloud
bias sign in most (85%) of the cloud top pressure and optical thickness bins. Therefore,
removing the "too thick bias" by parameter tuning only is considered difficult in this model.
The overestimation of both the cloud amount and optical thickness ("too thick bias")
contributes to the negative SCRE bias. To illustrate the importance of the "too thick bias" for
the SCRE bias, we plot the relationship between the SCRE and low-top cloud amount in
Figure 6. Note that we selected data of low-top clouds, which are not overlapped by middle-
top or high-top clouds in the figure; hence, the SCRE is not affected by clouds other than the
low-top clouds, which prevail in the Hawaiian Trade Cumulus Region. The figure shows that
SCRE negatively increases as the low-top cloud amount increases in both the observation and
MIROC5-PPE. However, the MIROC5-PPE shows a negatively larger SCRE compared with
the observation. It is larger by ~30 W/m$^2$, even if the models have the same cloud amount as
the observation, which indicates that the optical thickness of low-top clouds is overestimated
in the MIROC5-PPE. The above-mentioned characteristics are common to all PPE members
and the observation is outside the range of the PPE. This again indicates that we cannot
remove the "too thick bias" by parameter tuning only.

# 5 Characteristics of different tuning parameters

The results presented so far illustrate the difficulties in removing the TOA radiation and cloud biases by parameter tuning. At the same time, however, we also learned that parameter tuning enables us to control the model biases to some extent, demonstrating its benefit for model development. For example, the global mean SCRE bias can be reduced by as much as 50% by tuning only (Figure 1c). To obtain the desired effects by parameter tuning, we need to understand the characteristics of different tuning parameters. Therefore, in the following, we briefly describe the regions in which the tuning parameters in Table 1 control the model biases, focusing on the CRE.

We calculated the regression coefficients of the CRE on different tuning parameters for each latitude–longitude grid point, referring to the 56 members of the PPE, and plotted the geographical distribution of the coefficients in Figures 7 and 8. In addition, we calculated the regression of the ISCCP cloud properties (cloud amount, cloud optical thickness, and cloud top pressure) on the tuning parameters. The results are shown in the Appendix Figures A1, A2, and A3. Note that the tuning parameters were normalized to the range of 0.0 to 1.0; thus, the coefficients indicate the responses of the CRE and clouds to increase in the tuning parameters from the minimum to the maximum values in Table 1.

The tuning parameters, which are especially effective in controlling the shortwave CRE, are wcbmax and albice; wcbmax and albice can change the SCRE by more than 10 W/m$^2$ over low-latitude oceans and the Arctic, respectively (Figures 7a,j).

The parameter wcbmax is the maximum cumulus updraft velocity at the cloud base. Increasing the parameter leads to an increase in the cloud amount over low-latitude oceans (Figure A1a), which would increase the shortwave reflection by clouds and contribute to the negative increase in the SCRE, as indicated by the blue colour in Figure 7a. Indeed, the geographical distribution of the changes in the cloud amount and SCRE are similar to each other, which is consistent with the above-mentioned argument (Figures A1a, 7a).

Albice is the albedo of ice and snow. Increasing the parameter leads to an increase in the clear-sky albedo in high latitudes covered with ice and snow, which also decreases the albedo contrast between the clear- and all-sky components. Because the SCRE is proportional to this albedo contrast, it approaches zero by definition. Indeed, the SCRE shows a positive increase in high latitudes, as indicated by the red colour in Figure 7j, which is consistent with the above-mentioned argument. In addition, increasing the albice leads to the decrease in cloud

amount and cloud optical thickness in the Arctic (Figures A1j, A2j), which is also consistent with the change in SCRE (Figure 7j).

We confirmed in Figures 2a and 3e that the parametric uncertainty of the SCRE bias is exceptionally large in the Arctic compared with other latitudes. In the Arctic, albice is the most effective parameter controlling the SCRE based on Figure 7. We therefore surmise that the large uncertainty in the SCRE bias is mainly caused by perturbing the albice.

In addition to the wcbmax and albice, other parameters, such as clmd, vicec, b1, alp1, and ucmin, have a considerable impact on the SCRE (Figures 7c,d,e,g,i). Tuning these parameters leads to changes in the SCRE, which are consistent with the changes in the cloud amount or cloud optical thickness or in both of them (Figures A1, A2). To reduce the negative SCRE bias in low-latitude oceans, as shown in Figure 3a, the tuning of wcbmax, clmd, vicec, and b1 would be effective. On the other hand, the impact of tuning precz0, faz1, and tnuw would be relatively small.

Focusing on the longwave CRE, we find that the most effective parameters are wcbmax and vicec; wcbmax and vicec can change the LCRE by more than 10 $W/m^2$ in low latitudes (Figures 8a,d).

Increasing the wcbmax leads to changes in the cloud top pressure, which decreases in tropical Africa, western tropical Pacific, and the South Pacific Convergence Zone, while it increases in the subtropics, especially around South and Southeast Asia (Figure A3a). The decrease (increase) in the cloud top pressure would lead to a decrease (increase) in the cloud top temperature and upward longwave radiation, which would contribute to the increase (decrease) in the greenhouse effect of clouds and the LCRE. The geographical distribution of the changes in the cloud top pressure and LCRE are similar to each other, which is consistent with the above-mentioned argument (Figures A3a,8a).

The vicec parameter is a factor for the icefall speed. Increasing the parameter causes the increase in the icefall speed, decrease in the cloud amount (Figure A1d), and increase in the cloud top pressure (Figure A3d). Such changes of the cloud properties would contribute to the decrease of the greenhouse effect of clouds, which is consistent with the decrease in LCRE, as shown in Figure 8d.

## 6 Discussion

The results of the present study have implications for the future development of MIROC. Parameter tuning has only a limited capability of controlling the SCRE biases over low-latitude oceans and the Southern Ocean in MIROC5. Therefore, modifying the model structure should be given a high priority to effectively alleviate the biases. The results underline the importance of improving parameterizations based on cloud process studies. On the other hand, the SCRE bias in the Arctic can be fully controlled by tuning the albedo of snow and ice in the current model structure. However, we expect that the albedo will be predicted or diagnosed with a more physically-based parameterization in the future rather than being specified as a tuning parameter, which would make the tuning of the SCRE more difficult.

Which part of the model structure is responsible for the SCRE biases in MIROC5? One possible factor is insufficient vertical mixing in the lower troposphere. In MIROC5, the overestimation of the low-top cloud amount over low-latitude oceans is accompanied by the dry bias in the free troposphere above the low-top clouds, suggesting that vertical mixing in the lower troposphere, such as that caused by shallow convection, is insufficient. In order to test the idea, we implemented a shallow convection parameterization to the MIROC5 AGCM following Park and Bretherton (2009). We did some parameter tuning after the implementation to ensure that TOA radiation is balanced as before the implementation. The results show that the implementation (and the tuning) makes the SCRE more positive in low latitude oceans, which alleviates the negative SCRE bias (Figures 3a and 9).

As an illustration, we focus on a grid point in the eastern tropical Pacific and look at the vertical profile of cloud condensate (liquid plus ice) and its tendency in Figure 10. We find a large maximum of cloud condensate at 850hPa before the implementation of the shallow convection scheme (solid line in Figure 10a). This maximum is maintained by increasing tendencies from condensation, evaporation, turbulent mixing, and convection (black and light blue lines in Figure 10b), and also by decreasing tendency from precipitation (magenta line in Figure 10b). After the implementation, those tendencies become smaller than before (Figure 10c), and the maximum of cloud condensate at 850hPa disappears (broken line in Figure 10a). There appears an increasing tendency from shallow convection at upper levels around 600-800hPa (orange line in Figure 10c), but this does not lead to large increase in cloud condensate. The obtained results are consistent with the view that vertical mixing induced by

shallow convection causes upward transport of total water in the lower troposphere, which dehydrates the low-cloud layer and decreases the low cloud condensate, thereby making the SCRE less negative.

   As a next step, a research concerning the impact of shallow convection on cloud feedback would also be useful. Previous studies indicate that simulated strength of convective mixing between the lower and middle tropical troposphere is related to cloud feedback and climate sensitivity in multi-model ensembles (Sherwood et al. 2014, Kamae et al. 2016). The results suggest that shallow convective mixing contributes to inter-model spread in climate sensitivity, which causes difficulty in assessing the impact of climate change. In order to test the hypothesis, a multi-model comparison is proposed in which climate feedback is estimated with shallow convection turned on and off in AGCMs. The comparison is called Selected Process On/Off Klima Inter-comparison Experiment (SPOOKIE) phase 2, which is under the framework of Cloud Feedback Model Inter-comparison Project (CFMIP, Webb et al. 2017). We expect that the SPOOKIE phase 2 will facilitate better understanding of the connection between shallow convection and cloud feedback.

   The present study also has implications for the inter-model difference in the CRE simulated by the CMIP5 MME. The SCRE and LCRE simulated by the CMIP5 MME show a large inter-model spread. The spread is larger than that in MIROC5-PPE; therefore, the observation data are within the range of the CMIP5 ensemble members for both the global mean and zonal mean values (Dolinar et al. 2015, Flato et al. 2013). This large spread in CMIP5 MME stems from the inter-model difference in both the model structure and specified parameter settings. The results of the present study indicate that specified parameter settings can explain only a small part of the inter-model spread in CMIP5 MME, suggesting that most of the spread is attributable to the difference in the model structure. This is consistent with the view that modifying the model structure is important to alleviate the biases in SCRE and LCRE.

   However, we should note that the results of the model evaluation presented here depend on the design of the PPE experiment. For example, we restricted the number of the perturbed parameters to 10 and that of the PPE members to 56 based on the amount of available computational resources. If we increased the number of the perturbed parameters and PPE members, the inter-model difference of the TOA radiation and cloud biases might be larger than that of the present study. The importance of the PPE design to obtain large inter-model spread is illustrated by Yamazaki et al. (2013) who conducted a PPE experiment with an

AOGCM, HadCM3. They perturbed 33 parameters to create 20000 members in the PPE experiment. Although they subsampled the PPE members so that the TOA radiation balance is close to the observation, as was done by Shiogama et al. (2012), they showed that the inter-model difference of the climate sensitivity is larger than that of MIROC5-PPE or CMIP MME.

The choice of the model used for the PPE experiment is another important factor. If we employed a model other than MIROC5, the biases in the TOA radiation and clouds would be notably different from what we presented. Klein et al. (2013) reported that the bias of having too many optically thick clouds has been reduced from CFMIP1 to CFMIP2 MME, with the best models having eliminated this bias. If we used a model with a very small bias in optically thick clouds, we might be able to change the sign of the bias by parameter tuning only. Therefore, the dominance of structure-oriented bias as illustrated by the MIROC5-PPE does not necessarily indicate unimportance of the parameter-oriented bias in general, as the latter is a function of the former.

Another issue is whether we should include models with a large TOA radiation imbalance in the PPE members. We did not include such models, assuming that TOA radiation must be balanced in the pre-industrial climate simulations. However, such models could also be included in the PPE if we applied flux adjustment at the sea surface to suppress climate drift, which might increase the parametric uncertainty of the biases compared with the present study. For example, Yamazaki et al. (2013) reported that the parametric uncertainty of the climate sensitivity increases by adopting models with a large TOA radiation imbalance in their PPE experiment using HadCM3 AOGCM. Collins et al. (2006) also conducted a PPE experiment using HadCM3 AOGCM with flux adjustment. They showed that the parametric uncertainty of the TOA shortwave radiation in the global and annual mean is ~20 W/m$^2$, which is much larger than the results in the present study.

If we did not adopt the SIS method in the MIROC5-PPE, namely, if we included PPE members with large TOA radiation imbalance by applying flux adjustment, how much larger would the inter-model spread become compared with this study? To address this issue, we estimated inter-model spread of the TOA net radiation in the MIROC5-PPE for two sets of ensemble members: (1) 5000 members created with Latin Hypercube Sampling, which include members with large TOA radiative imbalance, and (2) 56 members with small TOA radiative imbalance, which are selected with the SIS method from the 5000 members in (1). We estimated standard deviation for the two sets of ensemble members and the ratio of (1) to

(2) is 6.25 to 1.0. Therefore, inter-model spread of the TOA net radiation would be about 6
times larger if we did not adopt the SIS method. For the sake of argument, we now assume
that the 6-fold increase in the inter-model spread occurs not only to the net radiation but also
to the SCRE. In this case, observation data would be within the range of the PPE spread in the
global mean SCRE, in contrast to what we have seen in Figure 1c. However, as for the SCRE
over the subtropical oceans as seen in Figure 3a, the observation data would still be outside
the range of the PPE. The above arguments are consistent with Yokohata et al. (2012), who
evaluated the SCRE bias of PPE experiments under present climate conditions. They used
output of the PPEs conducted with multiple GCMs, some of which employed flux adjustment,
and find that the SCRE cooling bias over the subtropical oceans appears in almost all PPE
members.
**7   Conclusion**
To discuss how much of the biases in the TOA radiation and clouds can be removed by
parameter tuning in the present-day simulation with a climate model of the CMIP5 generation,
we used a low-resolution version of the MIROC5 AOGCM and compared the output of the
PPE experiment in the pre-industrial control setting with satellite observation data. We
evaluated the biases in the TOA radiation and clouds and quantified the parametric
uncertainty of the biases. We used the output of the PPE experiment without flux adjustment,
which is consistent with the experimental design of the CMIP5. The results indicate that
removing or changing the sign of the biases by parameter tuning only is difficult. Especially,
the cooling bias of the SCRE in low latitudes could not be removed, neither in the zonal mean
nor at each latitude–longitude grid point. The bias was related to the overestimation of both
the cloud amount and cloud optical thickness, which could not be removed by parameter
tuning either. However, they could be alleviated by tuning parameters such as the maximum
cumulus updraft velocity at the cloud base. On the other hand, the bias of the SCRE in the
Arctic was sensitive to parameter tuning. It could be removed by tuning parameters such as
the albedo of ice and snow both in the zonal mean and at each grid point.
As discussed in Section 6, the obtained results of the PPE experiment are dependent on  the
model and experimental design. Especially, inter-model spread of the PPE is affected by
employing the SIS method. Whether the results are applicable to other models or PPE
experiments remains to be investigated further. However, the present study illustrates the

benefit of PPE experiments, which provide useful information regarding the model development strategy, namely, the effectiveness and limitations of parameter tuning. Based on the results of the present study, a parameterization for shallow convection was implemented in MIROC6 to alleviate the cloud bias over low-latitude oceans. Conducting PPE experiments with the future versions of MIROC is advisable to update our knowledge on the parametric uncertainty, which depends on the model structure; PPE experiments without flux adjustment using AOGCMs other than MIROC5 would also be useful to evaluate the biases in the simulated present climates, which are relevant for future projections in the CMIP5 MME.

## 8   Code and data availability

Source code of MIROC5, associated with this study is available to those who conduct collaborative research with the model users under licence from copyright holders. For further information on how to obtain the code please contact the corresponding author. The data from the model simulations and observations used in the analyses are available from the corresponding author upon request.

## Appendix A: Impact of parameter tuning on ISCCP cloud properties

The regression coefficients of the ISCCP cloud properties (cloud amount, cloud optical thickness, and cloud top pressure) on tuning parameters are shown here to help readers interpret the CRE changes in Figures 7 and 8.

## Acknowledgements

The authors thank Hideaki Kawai for valuable discussion and comments. The authors also thank Editage (www.editage.jp) for English language editing. This work was supported by the Program for Risk Information on Climate Change and the Integrated Research Program for Advancing Climate Models of the Ministry of Education, Culture, Sports, Science and Technology (MEXT), Japan. HS was supported by Grant-in-Aid 26281013 from the MEXT of Japan. The Earth Simulator at JAMSTEC and NEC SX at NIES were used to perform the model simulations.

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

Table 1. List of physics parameters that were varied in the MIROC5-PPE.

| Name | Category | Description | Standard | Min | Max |
|------|----------|-------------|----------|-----|-----|
| wcbmax[a] | Cumulus | Maximum cumulus updraft velocity at cloud base (m/s) | 1.7 | 0.7 | 2.8 |
| precz0[a] | Cumulus | Base height for cumulus precipitation (m) | 500 | 200 | 1000 |
| clmd[a] | Cumulus | Entrainment efficiency (ND) | 0.51 | 0.4 | 0.6 |
| vicec[b] | Cloud | Factor for ice falling speed ($m^{0.474}$/s) | 38 | 25 | 40 |
| b1[c] | Cloud | Berry parameter ($m^3$/kg) | 0.09 | 0.07 | 0.11 |
| faz1[d] | Turbulence | Factor for PBL overshooting (ND) | 1.5 | 1 | 3 |
| alp1[d] | Turbulence | Factor for length scale $L_T$ (ND) | 0.23 | 0.16 | 0.3 |
| tnuw[c] | Aerosol | Timescale for nucleation (s) | 18000 | 14400 | 21600 |
| ucmin[c] | Aerosol | Minimum cloud droplet number (liquid) ($m^{-3}$) | $2.5 \times 10^7$ | $2.2 \times 10^7$ | $3.0 \times 10^7$ |
| alb[e] | Surface | Albedo of ice and snow[f] | Medium | Low | High |

[a] Chikira and Sugiyama (2010)

[b] Wilson and Ballard (1999)

[c] Takemura et al. (2005, 2009)

[d] Nakanishi and Niino (2004)

[e] Takata et al. (2003) and Watanabe et al. (2010)

[f] "alb" indicates a collection of eight parameters corresponding to the albedo of ice and snow over sea and land

Table 2. Observation data used for the model evaluation. All data are monthly means.

| Variable | Dataset | Period | References |
|----------|---------|--------|------------|
| Top-of-atmosphere radiative fluxes | CERES-EBAF (Edition 4.0) | March 2000–January 2017 | Loeb et al. (2009) |
| | ERBE-S9 | January 1985–December 1989 | Barkstrom (1984) |
| | ISCCP-FD | January 1986–December 1990 | Zhang et al. (2004) |
| Cloud fraction | GCM simulator-oriented ISCCP cloud product | July 1983–June 2008 | Pincus et al. (2012), Rossow et al. (1996) |
| | CALIPSO-GOCCP | June 2006–December 2010 | Chepfer et al. (2010) |

14 .

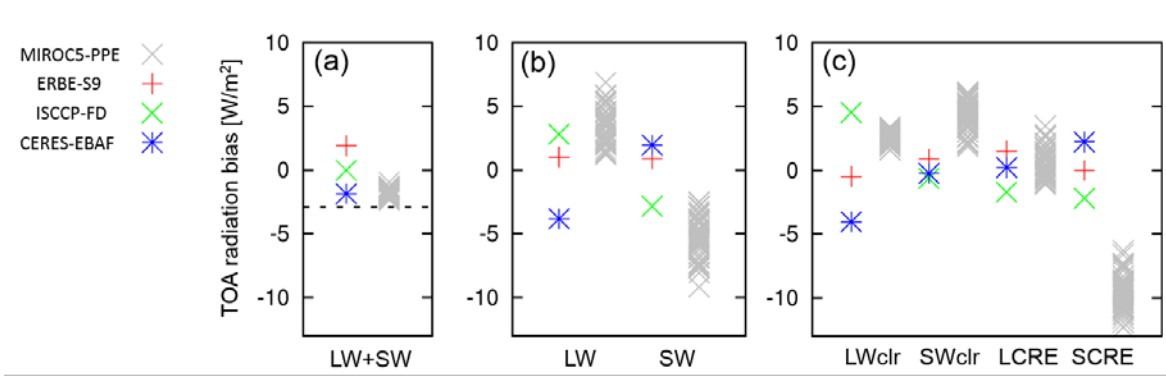

Figure 1.

TOA radiation bias in the global annual mean for (a) net, (b) longwave and shortwave, (c) longwave clear-sky, shortwave clear-sky, longwave CRE, and shortwave CRE components. The biases are with respect to the average of three observational data, namely, ERBE-S9, ISCCP-FD, and CERES-EBAF. The net radiation of zero with no TOA imbalance is indicated by the dashed line in (a). The unit is W/m$^2$ and the signs are positive downward.

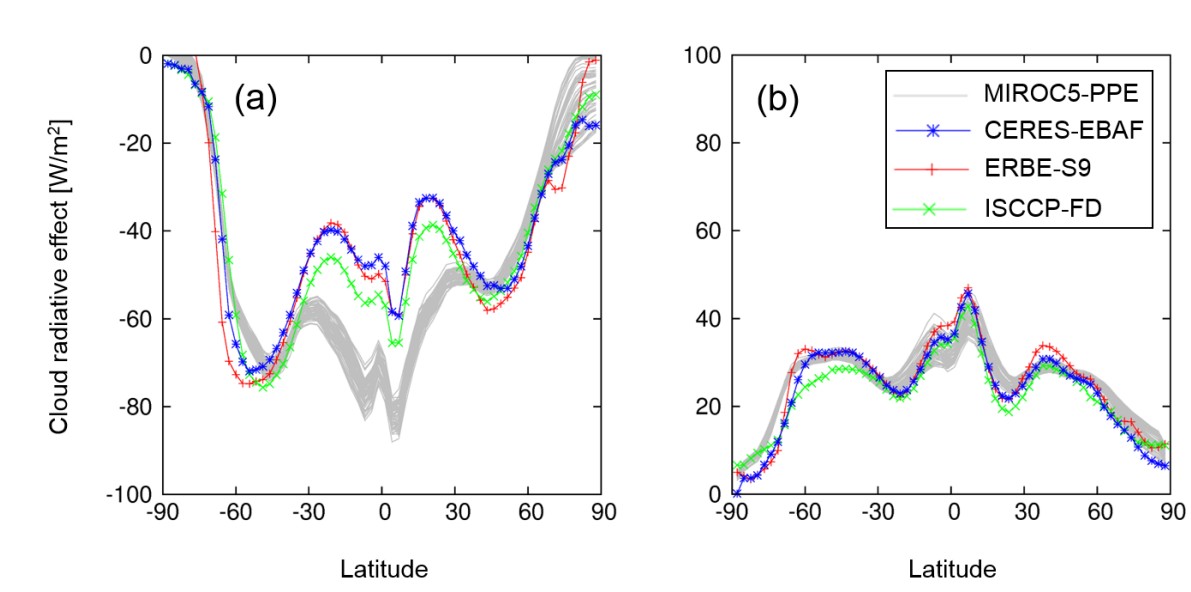

Figure 2.

TOA radiation in the zonal annual mean for the (a) shortwave CRE and (b) longwave CRE components. The unit is W/m² and the signs are positive downward.

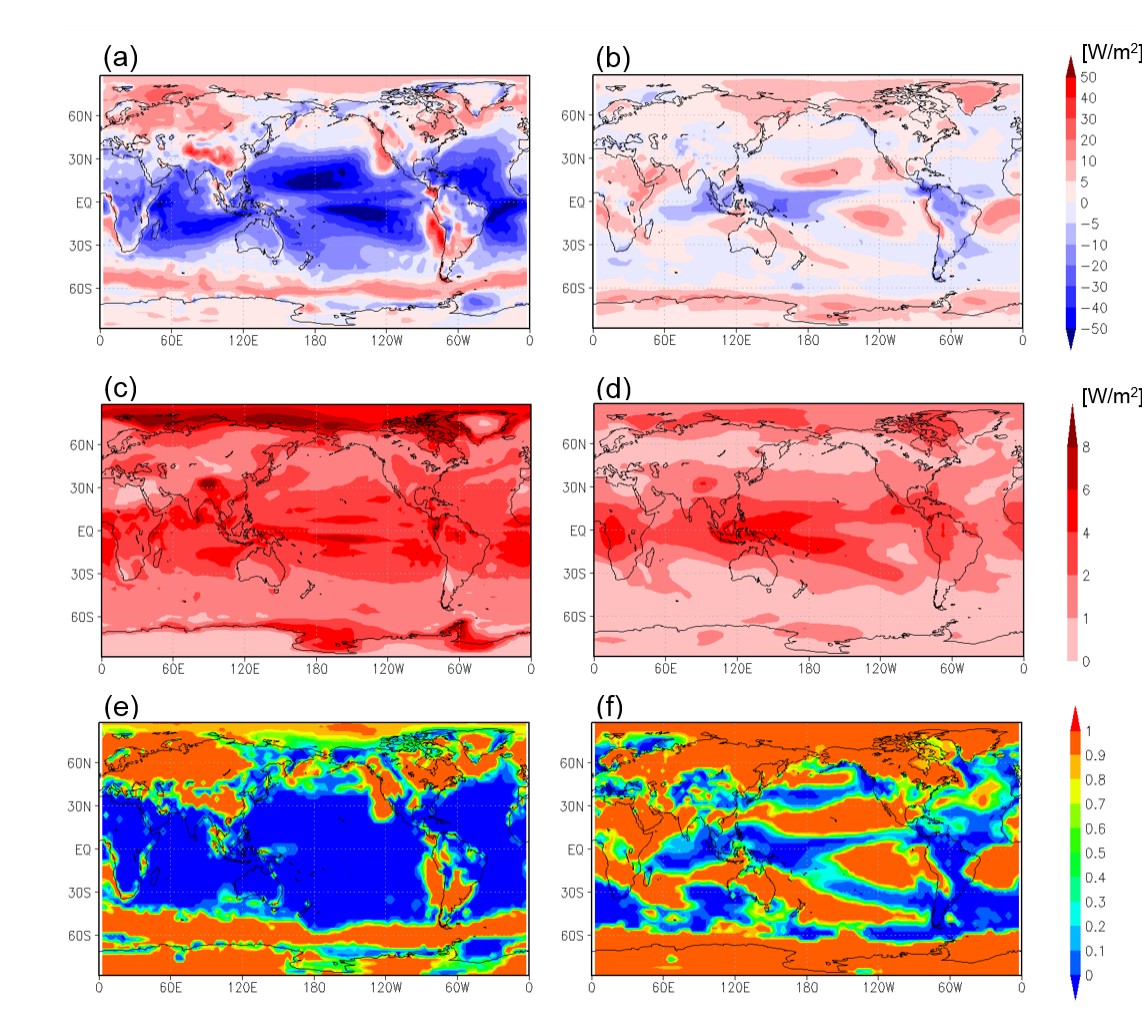

Figure 3.

TOA radiation bias in the annual mean for the (a) shortwave CRE and (b) longwave CRE components. The biases are for the ensemble mean of MIROC5-PPE with respect to CERES-EBAF. Standard deviation of the TOA radiation bias among the PPE ensemble members for the (c) shortwave CRE and (d) longwave CRE. Fraction of the PPE ensemble members, which have positive signs of the TOA radiation bias, for the (e) shortwave CRE and (f) longwave CRE.

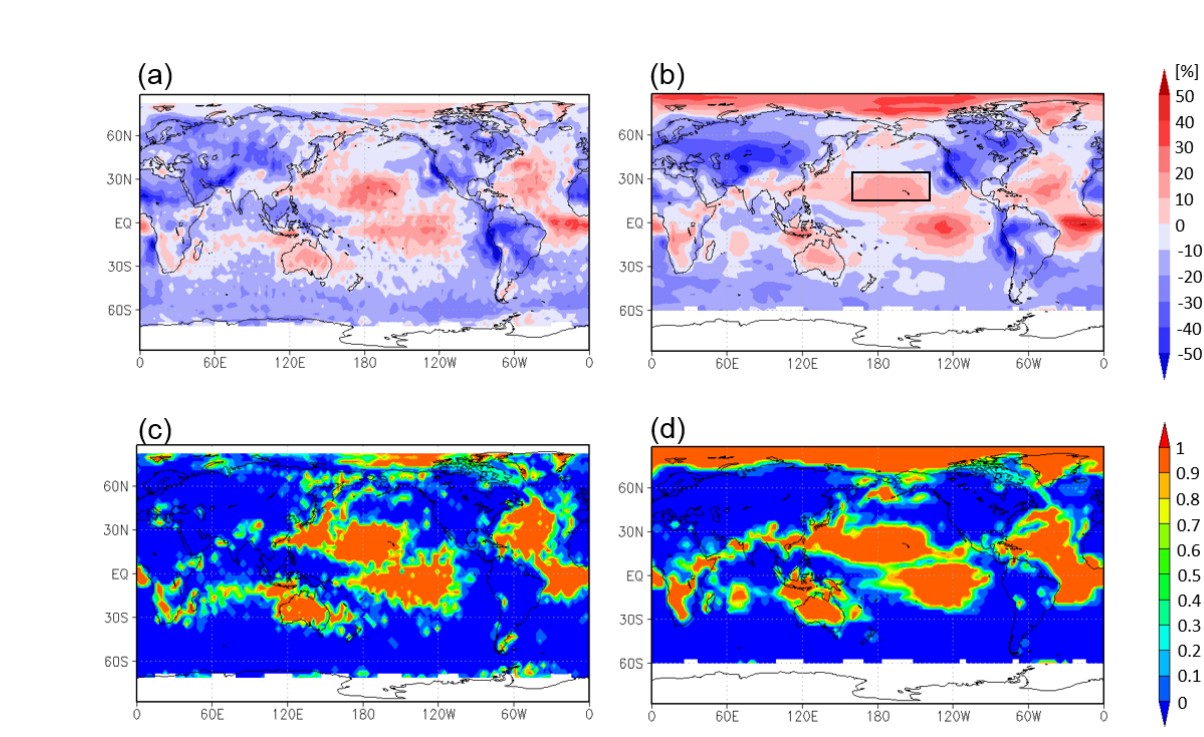

Figure 4.

Cloud amount bias in the July mean with respect to the (a) CALIPSO and (b) ISCCP observations. The biases are for the ensemble mean of MIROC5-PPE. Fraction of the PPE ensemble members, which have positive signs of the cloud amount bias, with respect to (c) CALIPSO and (d) ISCCP observation.

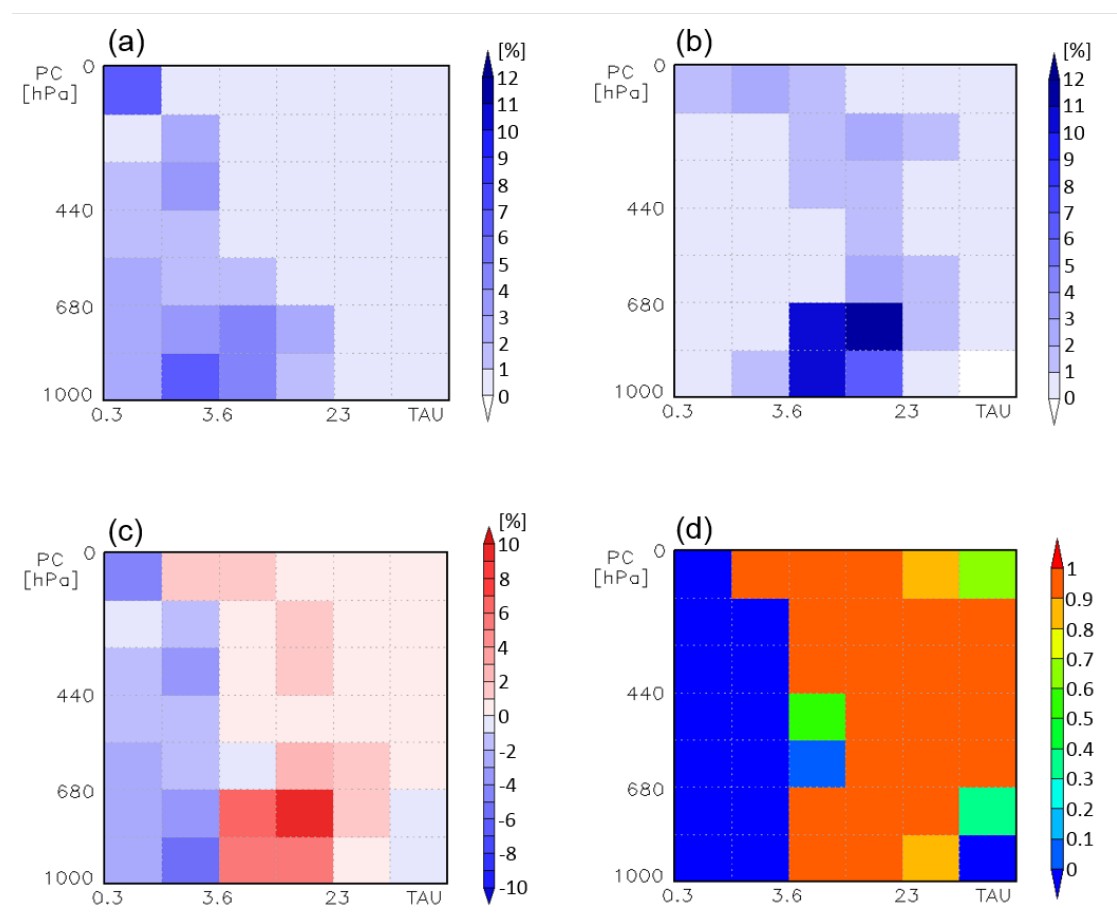

Figure 5.

ISCCP cloud amount of the July mean for the Hawaiian Trade Cumulus Region (15–35 °N, 160 °E–140 °W), indicated by the black square in Figure 4b, for different categories of the cloud top pressure (PC) and cloud optical thickness (TAU). Each panel is for (a) ISCCP observation, (b) MIROC5-PPE ensemble mean, (c) model bias, namely (b) minus (a), and (d) fraction of the PPE ensemble members with positive bias.

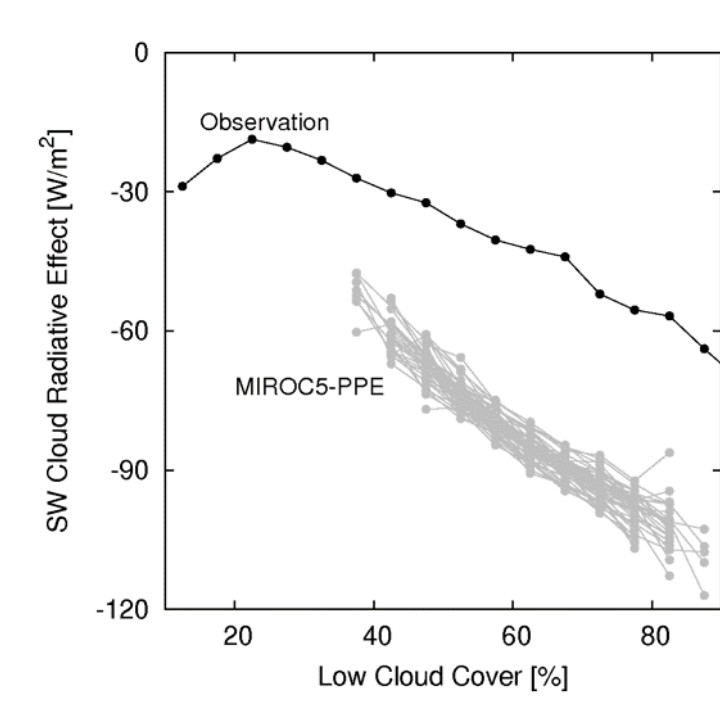

Figure 6.

Relationship between the non-overlapped low cloud amount and shortwave CRE of the July mean for the Hawaiian Trade Cumulus Region.

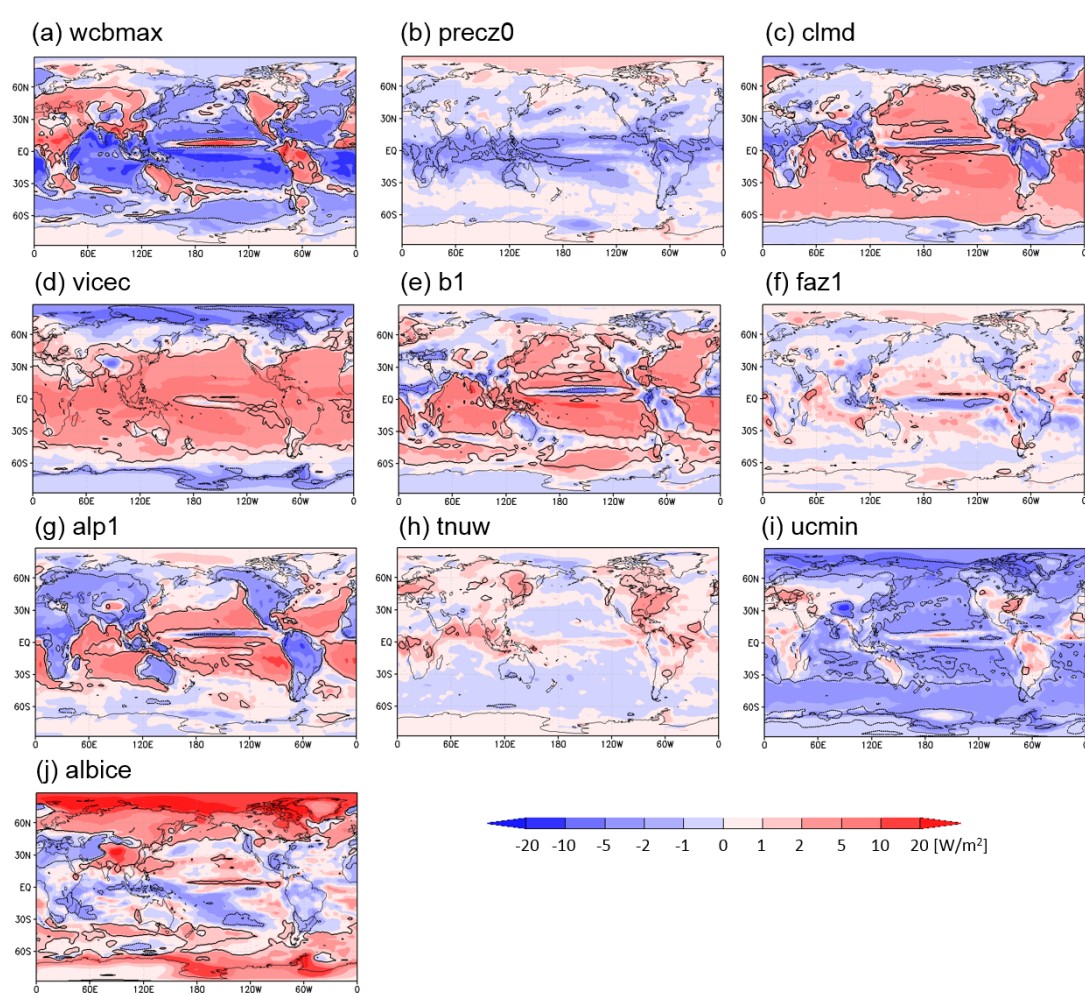

Figure 7.

Regression coefficient of the annual mean TOA shortwave CRE on the tuning parameters calculated with the 56 samples of the MIROC5-PPE. The definition of the tuning parameters is shown in Table 1. The tuning parameters are normalized to the range of [0,1]. The black curves indicate the threshold of the statistical significance with 5% level.

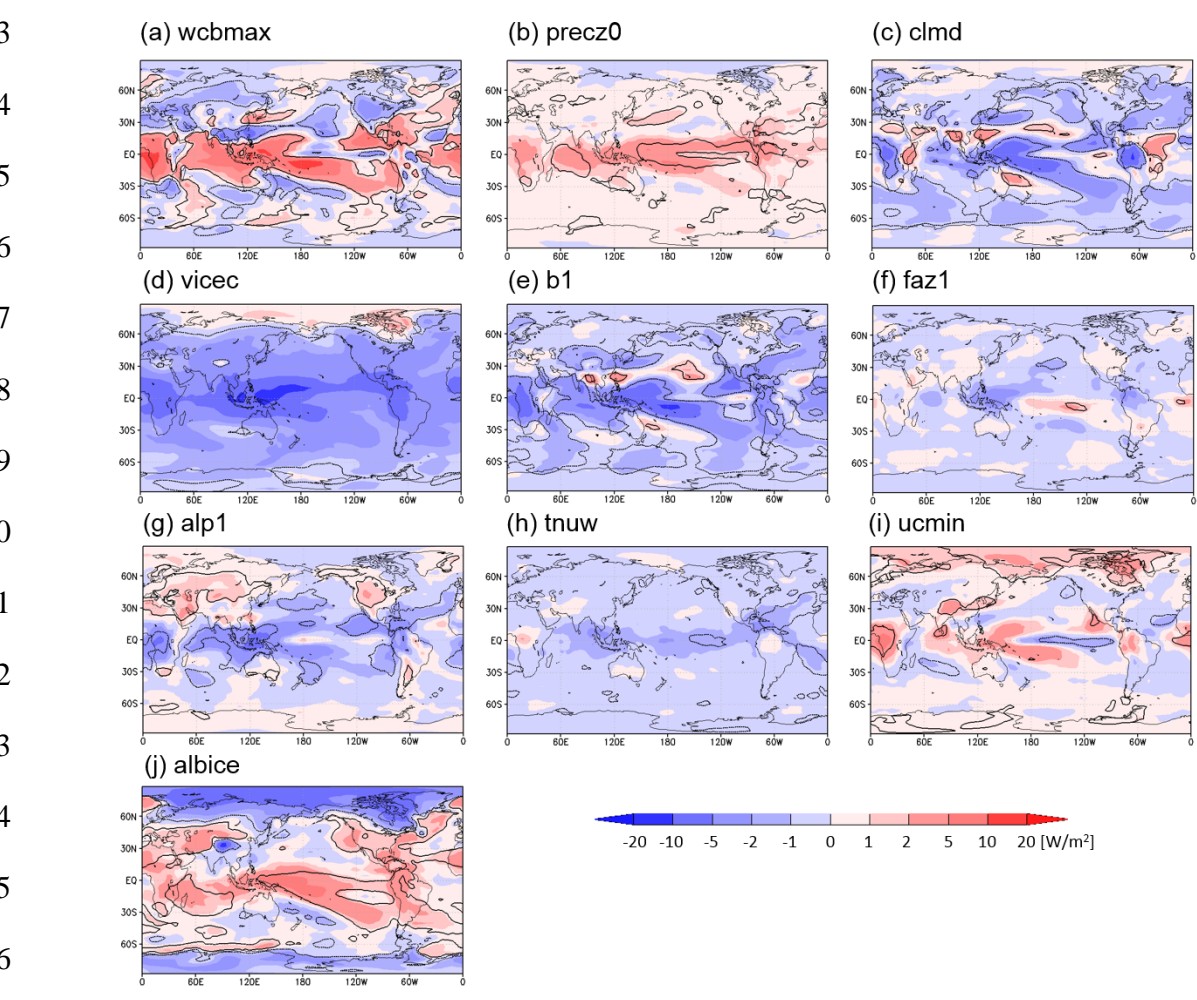

Figure 8.

Regression coefficient of the annual mean TOA longwave CRE on the tuning parameters calculated using the 56 samples of the MIROC5-PPE. The definition of the tuning parameters is shown in Table 1. The tuning parameters are normalized to the range of [0,1]. The black curves indicate the threshold of the statistical significance with 5% level.

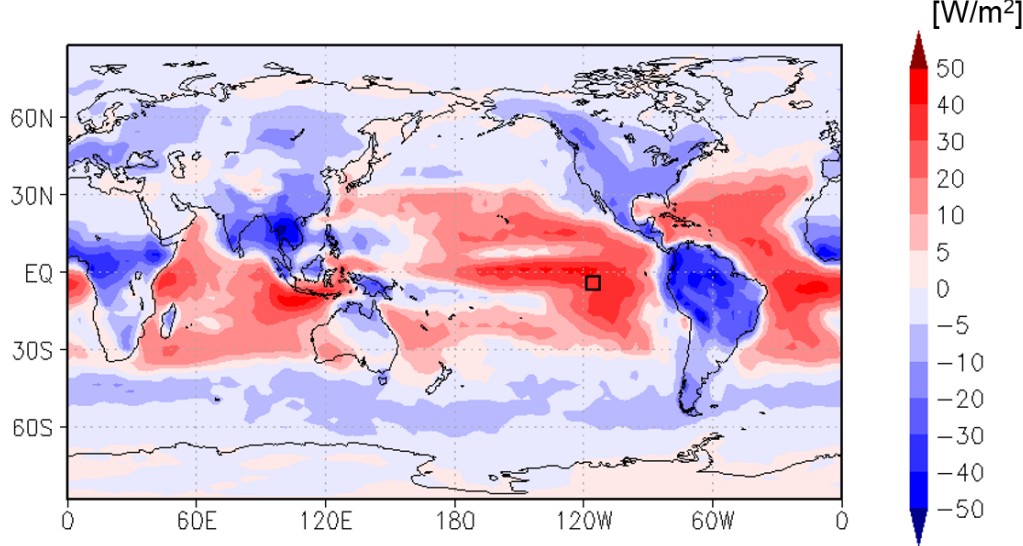

Figure 9.

Changes in annual mean TOA shortwave CRE induced by implementing a shallow

convection parameterization and parameter tuning in MIROC5 AGCM. The black

square in the eastern tropical Pacific indicates the position of a grid point focused in

Figure 10.

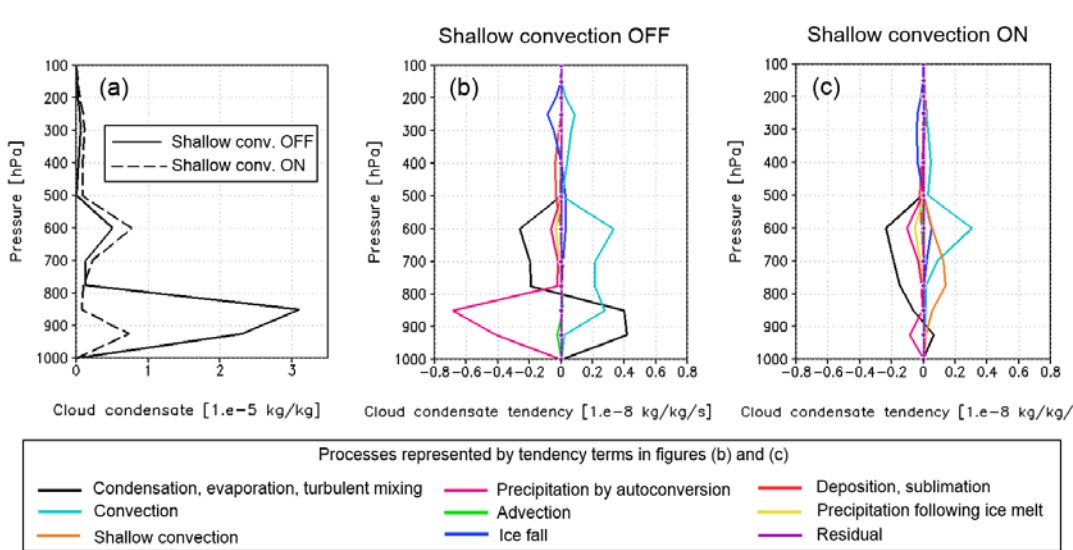

Figure 10.
Vertical profile of annual mean (a) cloud condensate and (b)(c) cloud condensate
tendencies in the eastern tropical Pacific simulated by MIROC5-AGCM. The data are
from the grid point located at $(114\,^\circ W, 5\,^\circ S)$, indicated by the black square in Figure
9. (a) cloud condensate simulated without shallow convection parameterization (solid
line) and with the parameterization (broken line), (b) cloud condensate tendencies
simulated without shallow convection parameterization, and (c) cloud condensate

tendencies simulated with the parameterization.

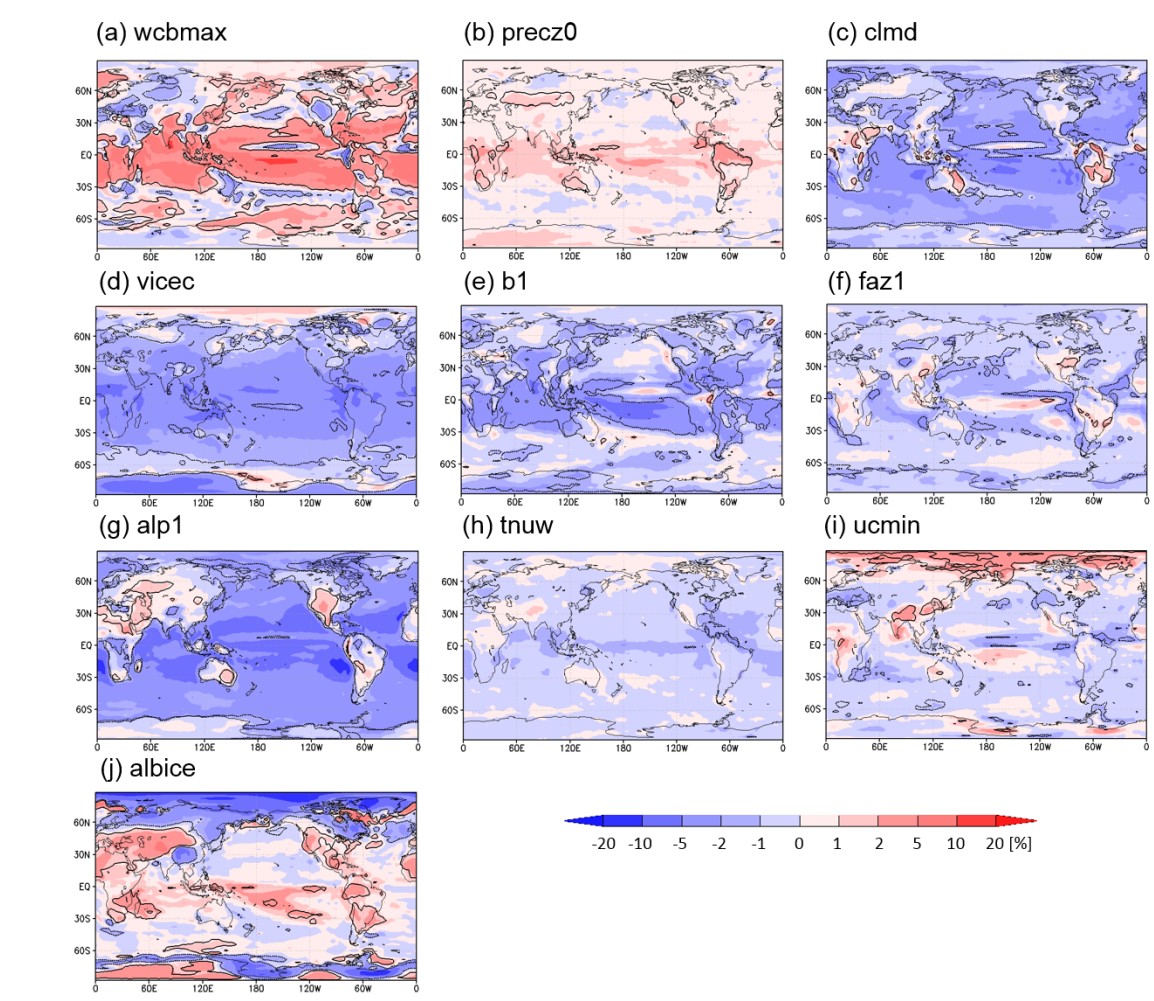

Figure A1.

Regression coefficient of the annual mean ISCCP cloud amount on the tuning parameters calculated using the 56 samples of the MIROC5-PPE. The definition of the tuning parameters is shown in Table 1. The tuning parameters are normalized to the range of [0,1]. The black curves indicate the threshold of the statistical significance with 5% level.

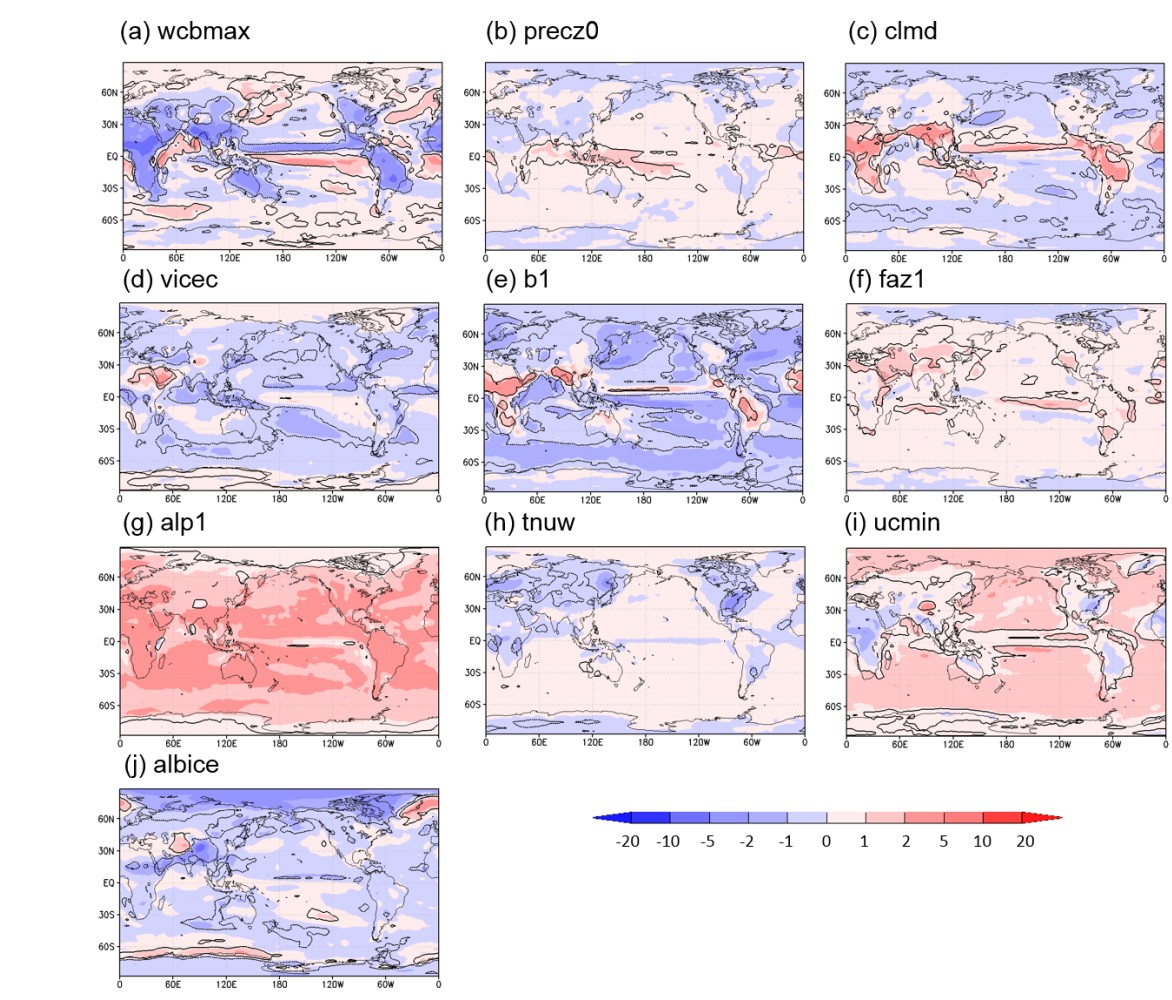

Figure A2.

Regression coefficient of the annual mean ISCCP cloud optical thickness on the tuning parameters calculated using the 56 samples of the MIROC5-PPE. The definition of the tuning parameters is shown in Table 1. The tuning parameters are normalized to the range of [0,1]. The black curves indicate the threshold of the statistical significance with 5% level.

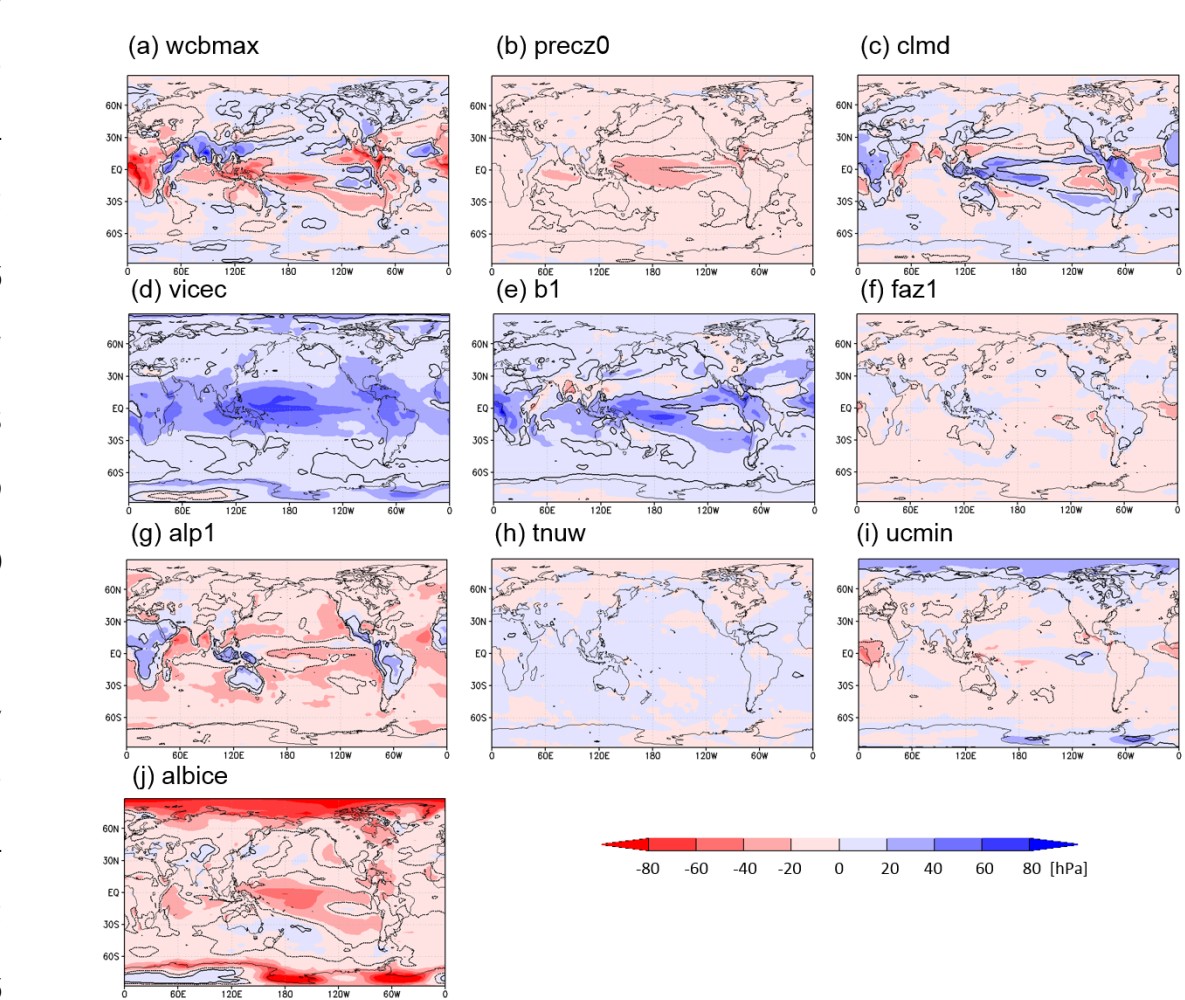

Figure A3.

Regression coefficient of the annual mean ISCCP cloud top pressure on the tuning parameters calculated using the 56 samples of the MIROC5-PPE. The definition of the tuning parameters is shown in Table 1. The tuning parameters are normalized to the range of [0,1]. The black curves indicate the threshold of the statistical significance with 5% level.