# Peer review of "Effectiveness and limitations of parameter tuning in reducing biases of top-of-atmosphere radiation and clouds in MIROC version 5"

_Geoscientific Model Development, 2017_

## Referee Comment (RC1) · Anonymous Referee #1 · 24 Jul 2017

PRINCIPAL CRITERIA : GMD Scientific significance: 1 Scientific quality: 1 Scientific reproducibility: 1 Presentation quality: 1

GENERAL COMMENTS:

Overall, this is a exceptionally well-written manuscript which discusses and quantifies the extent which model tuning influences long-standing biases in the TOA radiation budget. This work clearly demonstrates that 'model tuning' alone can not remove or change the sign of long-standing cloud and TOA radiative biases. The results also demonstrate the importance of an accurate representation physical processes coupling clouds and its environment, in other words, having an appropriate parametriza-

tion. These results are of vital importance to the Geoscientific Model Development community, and beyond, given that biases and radiation and clouds can affect projections of climate sensitivity.

This work also sheds light onto how low-level clouds, both in the tropics and Arctic regions react differently to parameter tuning. The ideas and approach are clearly described and thus reproducible by others.

SPECIFIC COMMENTS: - Which version of CERES-EBAF was used in the analysis? Please update it to Edition4.0 which was released on March 7, 2017. - Please include a description of the cloud parametrization used in MIROC5. Since the conclusions specify that the parametrization itself is key, it would be nice to know which one (RH vs PDF scheme) is used and future efforts that will be considered (i.e. CFMIP SPOOKIE experiments). - Please elaborate upon the description/thought processes regarding 'fractions of PPE members with bias 0 and 1' (first discussed on page 9, lines 5). I struggled slightly to follow the argument entirely. - Why do you argue "removing 'Too thick bias' by parameter tuning is only considered difficult in THIS model"? Isn't this a problem for other CMIP5 models too?

TECHNICAL CORRECTIONS: - Figure 3 c & d: Please change the limits of the color bar and remove the contour lines.

---

## Referee Comment (RC2) · Anonymous Referee #2 · 26 Jul 2017

Review on "Effectiveness and limitations of parameter tuning in reducing biases of top-of-atmosphere radiation and clouds in MIROC version 5"

Tomoo Ogura et al.

**Overview**

The study discussed how much model biases could be removed through parameter tuning using output from MIROC5 AOGCM PPE simulations.  The paper is well written and organized. The results should provide valuable guidance to future MIROC developments.  To make the results more useful to other models, however, I feel more discussion on what model deficiencies in representing the physics that caused the model biases would be necessary for this study to have broader impact to the field.  I recommend a minor/moderate revision before this paper can be accepted for publication. My comments are listed below.

**Major comments**

1. The major conclusion from this study is that many model biases (specifically SCRE) cannot be removed through parameter tuning. To me, this is not something new or is hard to understand. It has been recognized in the field for a long time. To make the study more valuable, it would be nice if the authors add more discussion on what potential model deficiencies cause these model errors, particularly for those biases that are common in current climate models. The authors have touched the point a bit, but more discussions are needed.
2. As the authors also admitted, the conclusion is largely constrained by the way how the PPE members and PPE design are selected. Although the authors feel that "Whether the main conclusions in the present study are affected by the uncertainty in the PPE design is a subject of future studies" (P14, L29-30), some additional analyses of the PPE simulations should help better understand the model behavior and make their conclusion robust.

**Minor comments**

1. P1, L19-23, starting with "We used a low-resolution …". This sentence is confusing.
2. The last paragraph on P2. It's hard to believe that all model biases can be explained by factor (b) – parameter settings.
3. P4, L9: remove the sentence after "if".
4. P5, L19; Impact of using the Suppressed Imbalance Sampling (SIS) method on the study needs to be carefully discussed. To me, the use of SIS has largely limited model responses to the perturbed parameters.
5. P6, L2-12. The description of the PPE simulations is confusing. Please clarify. What do you mean "The TOA radiative imbalance of the 5000 sampels is

estimated using the output of a PPE separately conducted using the atmospheric component of MIROC5"?

6. How many year's run was made in the 5000 samples?
7. P6, L13-15. Were the 56 members run for present day conditions with the atmospheric component of MIROC5?
8. P7, L27-29: the reduction of the SCEF bias may increase biases in other fields.
9. P13 L22: "The spread is larger than that in MIROC5-PPE". This is primarily due to how the PPE results are sampled. Using the SIS method significantly narrow-down the spread.
10. P14, L19-20: "We did not …". This is not a good argument.
11. P14. The first sentence of the last paragraph. As I mentioned earlier, I believe that understanding the impact of the uncertainty in the PPE design is quite critical for this paper.

---

## Referee Comment (RC3) · Anonymous Referee #1 · 26 Jul 2017

I've been thinking about how one could like the results of this paper with physical properties of shallow convection. One idea, which would also address RC2 comments, is plotting the tendencies of total water as in Figure 6 from Zhag et al, 2013.: CGILS Results on Low Cloud feedback.

---

## Short Comment (SC1) · 27 Jul 2017

Thank you for your suggestion. Plotting the tendencies would help to better understand how the the low cloud bias is maintained in MIROC5, and how the bias is affected by shallow convection. It would also help to address the RC2 comments. We will add some figures showing the tendencies and discussion in the revised manuscript.

---

## Author Comment (AC1) · 26 Oct 2017

In the following, we present responses to the comments by anonymous referees #1 and #2.

——————- Responses to the referee #1 start ——————-

Thank you for your comments concerning our manuscript submitted to the GMD. We found the comments most helpful and have revised the manuscript accordingly. The point-by-point listing of our response/action for each of your comments /suggestions is as follows. The line numbers are indicated for the revised version of the manuscript, while the numbers for the old version are also shown in parentheses.

Comment 1. Which version of CERES-EBAF was used in the analysis? Please update it to Edition4.0 which was released on March 7, 2017.

Response 1. We used Edition1A in the manuscript submitted to the GMD. Following your suggestion, we updated it to Edition4.0.

Chages in manuscript 1. We updated Figures 1, 2, 3, and 6 using Edition4.0 data. We also noted in Table 2 that we used Edition4.0 of CERES-EBAF.

Comment 2. Please include a description of the cloud parametrization used in MIROC5. Since the conclusions specify that the parametrization itself is key, it would be nice to know which one (RH vs PDF scheme) is used and future efforts that will be considered (i.e. CFMIP SPOOKIE experiments).

Response 2. We included a description of the cloud parametrization used in MIROC5. We also added discussion on future efforts that will be considered, including the CFMIP SPOOKIE experiments.

Changes in manuscript 2. We added a description at Page 5, Line 19 (Page 5, Line 9) as follows.

"The cloud parameterization of MIROC5 employs a statistical scheme. We assume that there is small-scale fluctuation of total water Qt within the model grid box, which is described by a probability density function (PDF), G(Qt). We also assume that the Qt exceeding supersaturation with respect to liquid , Qs, takes the form of cloud liquid. Then the cloud cover C and cloud liquid content Qc are diagnosed as the integral over the saturated part of the grid box, as follows:

Equations (1) and (2), see text for detail.

Overbar denotes average over the grid box. The shape of the PDF is represented by a triangular function. The model predicts variance and skewness of the PDF, which are affected by cumulus convection, cloud microphysics, turbulent mixing, and advection. Details of the cloud parameterization are described by Watanabe et al. (2009). MIROC5 also uses a cloud microphysics parameterization following Wilson and Ballard (1999). The parameterization predicts ice water content using physically-based tendency terms which represents nucleation, deposition and sublimation, riming, and ice melting, among others."

We added discussion at Page 15, Line 25 (Page 13, Line 30) as follows.

"As a next step, a research concerning the impact of shallow convection on cloud feedback would also be useful. Previous studies indicate that simulated strength of convective mixing between the lower and middle tropical troposphere is related to cloud feedback and climate sensitivity in multi-model ensembles (Sherwood et al. 2014, Kamae et al. 2016). The results suggest that shallow convective mixing contributes to inter-model spread in climate sensitivity, which causes difficulty in assessing the impact of climate change. In order to test the hypothesis, a multi-model comparison is proposed in which climate feedback is estimated with shallow convection turned on and off in AGCMs. The comparison is called Selected Process On/Off Klima Intercomparison Experiment (SPOOKIE) phase 2, which is under the framework of Cloud Feedback Model Intercomparison Project (CFMIP, Webb et al. 2017). We expect that the SPOOKIE phase 2 will facilitate better understanding of the connection between shallow convection and cloud feedback."

Comment 3. Please elaborate upon the description/thought processes regarding 'fractions of PPE members with bias 0 and 1' (first discussed on page 9, line 5). I struggled slightly to follow the argument entirely.

Response 3. We added description regarding the 'fraction of the PPE ensemble members which have positive signs of the TOA radiation bias'.

Changes in manuscript 3. We added description at Page 10, Line 15 (Page 9, Line 5) as follows.

"At each grid point, we count the number of the PPE members which have positive SCRE bias. Then we divide it by the total number of the PPE members, which is 56. The resulting fractions are plotted in the Figure 3e, so that we can see if the observation data lie within the range of the PPE spread at each grid point. In most areas of the globe, the fraction is 0 (blue ) or 1 (orange), which means that observation data are outside the range of the PPE spread, or all PPE members have the same sign of the SCRE bias."

Comment 4. (P.10, L14) Why do you argue "removing 'Too thick bias' by parameter tuning is only considered difficult in THIS model"? Isn't this a problem for other CMIP5 models too?

Response 4. Yes, we agree that the 'too thick bias' is a problem for most of other CMIP5 models, too. However, we have not confirmed if we can remove the bias by parameter tuning in other CMIP5 models. In order to confirm that, we need to check the output of PPE experiment conducted with other CMIP5 models, which are not available yet. That is why we argue that removing the bias by parameter tuning is only considered difficult in MIROC5 , which we confirmed by analyzing the PPE output in this study.

Comment 5. Figure 3c & d: Please change the limits of the color bar and remove the contour lines.

Response 5. We changed the limits of the color bar and removed the contour lines, so that readers can see the magnitude of the plotted variables only by the color shading.

Changes in manuscript 5. We updated the Figures 3c & d.

Comment 6. I've been thinking about how one could like the results of this paper with physical properties of shallow convection. One idea, which would also address RC2 comments, is plotting the tendencies of total water as in Figure 6 from Zhang et al, 2013.: CGILS Results on Low Cloud Feedback.

Response 6. Thank you for your suggestion. We plotted the tendencies of cloud condensate (liquid plus ice) in a similar way as the Figure 6 from Zhang et al. 2013. The tendencies help to understand how the vertical profile of cloud condensate is modified by different processes in AGCMs.

Changes in manuscript 6. We added figures for cloud condensate tendencies in Figures 10b,c.

————- Responses to the referee #1 end —————-

————- Responses to the referee #2 start —————-

Thank you for your comments concerning our manuscript submitted to the GMD. We found the comments most helpful and have revised the manuscript accordingly. The point-by-point listing of our response/action for each of your comments /suggestions is as follows. The line numbers are indicated for the revised version of the manuscript, while the numbers for the old version are also shown in parentheses.

Comment 1. The major conclusion from this study is that many model biases (specifically SCRE) cannot be removed through parameter tuning. To me, this is not something new or is hard to understand. It has been recognized in the field for a long time. To make the study more valuable, it would be nice if the authors add more discussion on what potential model deficiencies cause these model errors, particularly for those biases that are common in current climate models. The authors have touched the point a bit, but more discussions are needed.

Response 1. Following your suggestion, we added more discussion on what potential model deficiencies cause the SCRE biases over the low-latitude oceans. The SCRE biases are common in current climate models.

Changes in manuscript 1. We added discussion at Page 15, Line 1 (Page 13, Line 30) as follows.

"Which part of the model structure is responsible for the SCRE biases in MIROC5? One possible factor is insufficient vertical mixing in the lower troposphere. In MIROC5, the overestimation of the low-top cloud amount over low-latitude oceans is accompanied by the dry bias in the free troposphere above the low-top clouds, suggesting that vertical mixing in the lower troposphere, such as that caused by shallow convection, is insufficient. In order to test the idea, we implemented a shallow convection parameterization to the MIROC5 AGCM following Park and Bretherton (2009). We did some parameter tuning after the implementation to ensure that TOA radiation is balanced as before the implementation. The results show that the implementation (and the tuning) makes the SCRE more positive in low latitude oceans, which alleviates the negative SCRE bias (Figures 3a and 9). As an illustration, we focus on a grid point in the eastern tropical Pacific and look at the vertical profile of cloud condensate (liquid plus ice) and its tendency in Figure 10. We find a large maximum of cloud condensate at 850hPa before the implementation of the shallow convection scheme (solid line in Figure 10a). This maximum is maintained by increasing tendencies from condensation, evaporation, turbulent mixing, and convection (black and light blue lines in Figure 10b), and also by decreasing tendency from precipitation (magenta line in Figure 10b). After the implementation, those tendencies become smaller than before (Figure 10c), and the maximum of cloud condensate at 850hPa disappears (broken line in Figure 10a). There appears an increasing tendency from shallow convection at upper levels around 600-800hPa (orange line in Figure 10c), but this does not lead to large increase in cloud condensate. The obtained results are consistent with the view that vertical mixing induced by shallow convection causes upward trasnport of total water in the lower troposphere, which dehydrates the low-cloud layer and decreases the low cloud condensate, thereby making the SCRE less negative."

Comment 2. As the authors also admitted, the conclusion is largely constrained by the way how the PPE members and PPE design are selected. Although the authors feel that "Whether the main conclusions in the present study are affected by the uncertainty in the PPE design is a subject of future studies" (P14, L29-30), some additional analyses of the PPE simulations should help better understand the model behavior and make their conclusion robust.

Response 2. Following your suggestion, we added some analyses of the PPE simulations and discussed whether the obtained results are robust with respect to the PPE design.

Changes in manuscript 2. We added discussion at Page 17, Line 15 (Page 15, Line 7) as follows.

"If we did not adopt the SIS method in the MIROC5-PPE, namely, if we included PPE members with large TOA radiation imbalance by applying flux adjustment, how much larger would the inter-model spread become compared with this study? To address this issue, we estimated inter-model spread of the TOA net radiation in the MIROC5-PPE for two sets of ensemble members: (1) 5000 members created with Latin Hypercube Sampling, which include members with large TOA radiative imbalance, and (2) 56 members with small TOA radiative imbalance, which are selected with the SIS method from the 5000 members in (1). We estimated standard deviation for the two sets of ensemble members and the ratio of (1) to (2) is 6.25 to 1.0. Therefore, inter-model spread of the TOA net radiation would be about 6 times larger if we did not adopt the SIS method. For the sake of argument, we now assume that the 6-fold increase in the inter-model spread occurs not only to the net radiation but also to the SCRE. In this case, observation data would be within the range of the PPE spread in the global mean SCRE, in contrast to what we have seen in Figure 1c. However, as for the SCRE over the subtropical oceans as seen in Figure 3a, the observation data would still be outside the range of the PPE. The above arguments are consistent with Yokohata et al. (2012), who evaluated the SCRE bias of PPE experiments under present climate conditions. They used output of the PPEs conducted with multiple GCMs, some of which employed flux adjustment, and find that the SCRE cooling bias over the subtropical oceans appears in almost all PPE members."

Comment 3. P1, L19-23, starting with "We used a low-resolution ...". This sentence is confusing.

Response 3. We rearranged the sentence to make the message clear.

Changes in manuscript 3. We rearranged the sentences at Page 1, Line 21 (Page 1, Line 19) as follows.

"We used output of a perturbed parameter ensemble (PPE) experiment conducted with an Atmosphere-Ocean General Circulation Model (AOGCM) without flux adjustment. The Model for Interdisciplinary Research on Climate version 5 (MIROC5) was used for the PPE experiment. Output of the PPE was compared with satellite observation data to evaluate the model biases and the parametric uncertainty of the biases with respect to TOA radiation and clouds."

Comment 4. The last paragraph on P2. It's hard to believe that all model biases can be explained by factor (b) - parameter settings.

Response 4. We agree with your comment, namely, it is not likely that all kinds of biases are completely explained by factor (b) - parameter settings. However, the last paragraph on P2 is meant to argue something different from above. Rather it argues that some specific biases in question may be completely explained by factor (b). For example, Figure 2(a) shows that the zonal mean SCRE biases in the Arctic can be completely explained by the factor (b). We rearranged the paragraph to make the message clear.

Changes in manuscript 4. We rearranged the sentences at Page 3, Line 10 (Page 3, Line 1) as follows.

"if the biases in question can be completely explained by factor (b)"

Comment 5. P4, L9:remove the sentence after "if".

Response 5. Following your suggestion, we removed the sentence after "if".

Changes in manuscript 5. We removed the following sentence at Page 4, Line 18 (Page 4, Line 9).

[Figure]

"if the model-generated clouds existed in the real world"

Comment 6. P5, L19; Impact of using the Suppressed Imbalance Sampling (SIS) method on the study needs to be carefully discussed. To me, the use of SIS has largely limited model responses to the perturbed parameters.

Response 6. We added discussion regarding the impact of using the Suppressed Imbalance Sampling (SIS) method on the study. We estimate that inter-model spread in TOA net radiation becomes 1/6.25, namely 16% of the original by adopting the SIS method. For more detail, please refer to "Changes in manuscript 2".

Changes in manuscript 6. Please refer to "Changes in manuscript 2".

Comment 7. P6, L2-12. The description of the PPE simulations is confusing. Please clarify. What do you mean "The TOA radiative imbalance of the 5000 samples is estimated using the output of a PPE separately conducted using the atmospheric component of MIROC5"?

Response 7. For clarification, we added some description of how we estimated the TOA radiative imbalance of the 5000 samples.

Changes in manuscript 7. We added the following sentences at Page 7, Line 6 (Page 6, Line 6).

"The selection of the 56 members is conducted with the following 3 steps: 1) we conduct a PPE experiment with MIROC5 AGCM under pre-industrial condition, in which tuning parameters are changed one at a time to the minimum and maximum values before running the AGCM for 6 years, 2) output of the PPE members are linearly interpolated to estimate the TOA radiative imbalance for the 5000 samples of the tuning parameters, and finally, 3) we select 56 members in which the TOA radiative imbalance is close to that of the standard model."

Comment 8. How many year's run was made in the 5000 samples?

Response 8. We did not run the AGCM or the AOGCM in the 5000 samples. The TOA radiative imbalance in the 5000 samples is estimated by linearly interpolating the output of the PPE experiment conducted with the AGCM. The AGCM is integrated for 6 years in 43 samples of tuning parameters to construct the PPE output.

Changes in manuscript 8. Please refer to "Changes in manuscript 7".

Comment 9. P6, L13-15. Were the 56 members run for present day conditions with the atmospheric component of MIROC5?

Response 9. No, the 56 members were run with the MIROC5 AOGCM. We changed the sentence to make the message clear.

Changes in manuscript 9. At Page 7, Line 19(Page 6, Line 13), we changed the sentence from "We ran the 56 members of the model" to "We ran the 56 members of the MIROC5 AOGCM".

Comment 10. P7, L27-29:the reduction of the SCRE bias may increase biases in other fields.

Response 10. This is an important issue. Thank you for pointing this out. We added a summary for the correlation between the SCRE bias and other components of the radiation bias.

Changes in manuscript 10. We added the following sentences at Page 9, Line 16(Page 8, Line 10).

"We should note that the SCRE biases are negatively correlated with the LCRE biases with the correlation coefficient of -0.82. Therefore, if we reduce the SCRE bias by making it more positive, the LCRE bias tends to be more negative. This would reduce the LCRE bias in more than half of the PPE members. Correlations of the SCRE biases with the biases in clear sky components are small: -0.08 with LWclr and -0.32 with SWclr."

Comment 11. P13 L22: "The spread is larger than that in MIROC5-PPE". This is primarily due to how the PPE results are sampled. Using the SIS method significantly narrow-down the spread.

Response 11. We added discussion regarding the impact of using the Suppressed Imbalance Sampling (SIS) method on the study. We estimate that inter-model spread in TOA net radiation becomes 1/6.25, namely 16% of the original by adopting the SIS method. For more detail, please refer to "Changes in manuscript 2".

Changes in manuscript 11. Please refer to "Changes in manuscript 2".

Comment 12. P14, L19-20: "We did not ...". This is not a good argument.

Response 12. We decided not to include models with large TOA radiative imbalance in the PPE. This decision is based on the assumption that TOA radiation must be balanced in the pre-industrial climate simulations. We rearranged the sentence to make this assumption clear to the readers.

Changes in manuscript 12. We rearranged the sentences at Page 17, Line 4 (Page 14, Line 29), as follows.

"We did not include such models, assuming that TOA radiation must be balanced in the pre-industrial climate simulations."

Comment 13. P14. The first sentence of the last paragraph. As I mentioned earlier, I believe that understanding the impact of the uncertainty in the PPE design is quite critical for this paper.

Response 13. Following your suggestion, we added discussion regarding the impact of using the Suppressed Imbalance Sampling (SIS) method on the study. For more detail, please refer to "Changes in manuscript 2".

In addition, we rearranged the sentences at the beginning of the last paragraph to mention that inter-model spread of the PPE is affected by the SIS method.

Changes in manuscript 13. Please refer to "Changes in manuscript 2".

In addition, we rearranged the sentences at Page 19, Line 1 (Page 15, Line 17), as follows.

"As discussed in section 6, the obtained results of the PPE experiment are dependent on the model and experimental design. Especially, inter-model spread of the PPE is affected by employing the SIS method. Whether the results are applicable to other models or PPE experiments remains to be investigated further."

———————- Responses to the referee #2 end ——————-

Please also note the supplement to this comment:
https://www.geosci-model-dev-discuss.net/gmd-2017-117/gmd-2017-117-AC1-supplement.pdf

[Figure]

**Supplement:**

[revised manuscript text omitted]

~~Whether the main conclusions in the present study are affected by the uncertainty in the PPE design is a subject of future studies. Based on a previous study, however, we speculate that removing the SCRE cooling bias over subtropical oceans by parameter tuning only might be difficult, even if we increased the PPE members by applying flux adjustment. Yokohata et al. (2012) evaluated the SCRE bias of PPE experiments under present climate conditions using an AOGCM and two ASGCMs with flux adjustment and an AGCM with prescribed SST. They reported that the cooling bias appears over subtropical oceans in almost all PPE members. This result is consistent with the idea that Suppressed Imbalance Sampling adopted in the present study is not the only reason for the robustness of the SCRE cooling bias, which cannot be removed by parameter tuning.~~

[revised manuscript text omitted]